# The Health Effects of Chocolate and Cocoa: A Systematic Review

**DOI:** 10.3390/nu13092909

**Published:** 2021-08-24

**Authors:** Terence Yew Chin Tan, Xin Yi Lim, Julie Hsiao Hui Yeo, Shaun Wen Huey Lee, Nai Ming Lai

**Affiliations:** 1Herbal Medicine Research Centre, Institute for Medical Research, Ministry of Health, Setia Alam 40170, Malaysia; limxinyi.lim@gmail.com; 2Hospital Sultanah Bahiyah, Ministry of Health, Alor Setar 05460, Malaysia; julieyeohh@gmail.com; 3School of Pharmacy, Monash University Malaysia, Bandar Sunway 47500, Malaysia; shaun.lee@monash.edu (S.W.H.L.); lainm123@gmail.com (N.M.L.); 4School of Medicine, Taylor’s University, Subang Jaya 47100, Malaysia

**Keywords:** chocolate, cocoa, health benefits

## Abstract

Chocolate has a history of human consumption tracing back to 400 AD and is rich in polyphenols such as catechins, anthocyanidins, and pro anthocyanidins. As chocolate and cocoa product consumption, along with interest in them as functional foods, increases worldwide, there is a need to systematically and critically appraise the available clinical evidence on their health effects. A systematic search was conducted on electronic databases such as MEDLINE, EMBASE, and Cochrane Central Register of Controlled Trials (CENTRAL) using a search strategy and keywords. Among the many health effects assessed on several outcomes (including skin, cardiovascular, anthropometric, cognitive, and quality of life), we found that compared to controls, chocolate or cocoa product consumption significantly improved lipid profiles (triglycerides), while the effects of chocolate on all other outcome parameters were not significantly different. In conclusion, low-to-moderate-quality evidence with short duration of research (majority 4–6 weeks) showed no significant difference between the effects of chocolate and control groups on parameters related to skin, blood pressure, lipid profile, cognitive function, anthropometry, blood glucose, and quality of life regardless of form, dose, and duration among healthy individuals. It was generally well accepted by study subjects, with gastrointestinal disturbances and unpalatability being the most reported concerns.

## 1. Introduction

Chocolate has a long history of being consumed for its fine flavours as a luxury food since ancient times. The origins of chocolate can be traced back to 400 AD [1]. Chocolate is produced from cacao beans through a multistep process involving fermentation, drying, roasting, nib grinding and refining, conching, and tempering to ensure its stability and flavour [2,3]. The transformation process steps are first, fermentation of cacao beans to develop the chocolate flavour, followed by removal of water content by drying, then roasting, cleaning, and shelling of beans into nibs. Nibs are then ground and refined into cacao liquor, before being finally combined with various ingredients to produce different types of chocolate, such as dark chocolate, milk chocolate, and white chocolate [4]. The latin name for the cacao tree, *Theobroma cacao* L., means ‘Food of Gods’ [5]. Chocolate contains mostly fat (in the form of cacao butter) and is rich in polyphenols, such as catechins, anthocyanidins, and pro anthocyanidins [6]. The polyphenol content of chocolate varies with different raw ingredient sources and manufacturing processes [3,7].

The polyphenols of chocolate, which originates from cacao beans, are thought to partially contribute to the cardiometabolic health benefits of chocolate in modulating blood pressure and lipid profiles [8]. Several meta-analyses have suggested the benefits of chocolate consumption in reducing the risk of cardiometabolic events including coronary heart disease, myocardial infarction, stroke, and diabetes [9,10]. Apart from potential health promoting cardiometabolic effects, chocolate consumption has also been reported to improve cognition in clinical trials [11,12], supported by preclinical studies [13,14]. The cognitive benefits of chocolate are further supported by a recent systematic review which reported improvement in cognitive scores or task performance among young adults (less than 25 years old) and children with chronic chocolate consumption, again possibly attributed to polyphenols, including flavanols [15]. In addition to these effects, perhaps one of the most popular yet insufficiently substantiated health benefits or undesirable effects of chocolate consumption is its effect on skin health including aging and acne, which remains debatable [16]. Other less explored potential medicinal properties of chocolate include anticancer and antimicrobial effects [17,18,19]. Apart from polyphenols (flavonoids), other bioactive compounds of interest that are found in chocolate include methylxanthines such as theobromine and caffeine [20]. 

Based on the world’s average chocolate consumption which is estimated to be 0.9 kg per capita per year, there is no doubt that chocolate consumption continues to increase worldwide, along with growing interest in it as a functional food [21,22]. Therefore, there is a need to systematically and critically appraise the available clinical evidence of the health effects of chocolate. At present, published systematic reviews focus on analysing cardiometabolic effects of chocolate consumption among both patients (with pre-exisiting cardiometabolic co-morbidities) and healthy volunteers in single study populations. Some of the methodological limitations identified in these reviews include self-reporting of chocolate consumption and variation in intervention and comparators which include many types of chocolate [9,10]. To the best of our knowledge, there has been no systematic review done to assess the global health effects of chocolate or cocoa product consumption in the general healthy population. Hence, we are interested in formally assessing the quality of available evidence to enable better informed decisions at individual level or higher on the overall health effects of chocolate and cocoa product consumption. To improve the accuracy of analysis that health effects are attributable to chocolate or cocoa products, this systematic review analysed and presents the results of randomised clinical trials that assessed the health benefits of chocolate or cocoa product consumption in a healthy population, with a comparator group that did not consume any chocolate or cocoa products. 

## 2. Materials and Methods

### 2.1. Review Objective

This systematic review was conducted to determine the health effects of chocolate and cocoa product ingestion in healthy human subjects.

### 2.2. Inclusion and Exclusion Criteria

#### 2.2.1. Type of Study

This review considered all randomised controlled trials regardless of blinding, number or treatment arms, as well as both parallel-arm and cross-over designs.

#### 2.2.2. Type of Participants

This review included studies that recruited healthy human subjects, namely, those without a clinical diagnosis of any medical condition as participants.

#### 2.2.3. Type of Intervention

This review considered chocolate or cocoa product ingestion of any duration of four weeks or more, in any form, at any dose or frequency of doses, versus placebo, no intervention, or any other forms of non-cocoa related supplementation or intervention for comparison. We only included studies with a duration of intervention of equal or longer than four weeks to increase the robustness of our data analysis in addressing our study objective of adequately assessing the health effects of chocolate consumption.

To assess chocolate or cocoa products as a whole as well as a single contributing intervention, this review excluded studies with the following interventions:Any studies using only isolated cocoa constituents as intervention.Any studies on co-intervention in combination with chocolate or cocoa products.

#### 2.2.4. Type of Outcomes

To allow for a systematic assessment of outcome measures, the following primary and secondary outcomes were selected prior to screening and selection of studies. These outcomes were selected based on popular claims of potential health benefits found from a published literature review and general web search.

Primary outcomes

i.Effects on skin
PhotoprotectionPhotoagingii.Cardiovascular clinical outcomes
Risk of myocardial infarctionRisk of strokeIncidence of death due to cardiovascular event
iii.Cardiovascular parameters
(a)Changes in blood pressure
Systolic blood pressureDiastolic blood pressure
(b)Changes in lipid profile
Total cholesterol levelHDL levelLDL levelTriglyceride level(c)Blood glucose parameters
Fasting blood glucose
(d)Anthropometric parameters
WeightBMIOther potentially relevant quantifiable outcomes including waist circumference and body fat percentage



Secondary outcomes

(a)Cognitive outcomes in any validated measure
Overall cognitive functioningSpecific cognitive subdomain
MemoryReaction timeExecution

(b)Psychological outcomes in any validated measure
MoodDepressionAnxiety
(c)Effects on immunity(d)Anti-cancer effects(e)Quality of life(f)Adverse event (e.g., cravings, headache, allergy)

### 2.3. Search Strategy

Electronic databases MEDLINE, EMBASE, and Cochrane Central Register of Controlled Trials (CENTRAL) were searched for published studies, while trial registries including the WHO International Trial Registry Platform and ClinicalTrials.gov were searched for on-going studies since inception until March 2021. There were no restrictions applied in terms of publication period and language. In addition to database searches, the team retrieved the reference lists and citations of retrieved articles to further identify studies for inclusion, and contacted authors of relevant on-going trials to request details of any additional unpublished or ongoing studies that meet the inclusion criteria for this review. However, we did not receive any replies. The MEDLINE search strategies (Appendix A) were translated into the other databases using the appropriate controlled vocabulary as applicable.

The search terms used were chocolates or cocoa or cacao, and searches were limited to clinical studies. 

### 2.4. Study Selection

Two review authors independently screened titles and abstracts from the search strategy according to the inclusion and exclusion criteria, with disagreements resolved via discussion, with the help of a third author as an arbiter if required. The study selection process is outlined using the PRISMA diagram (Figure 1).

### 2.5. Data Extraction & Management

Two review authors coded all data from each included study independently using a pro forma designed specifically for this review. The interventions defined in the study were compared against our pre-defined intervention. Any disagreement among the review authors was resolved by discussion leading to a consensus, with referral to a third review author if necessary.

### 2.6. Data Analysis

#### 2.6.1. Risk of Bias Assessment

Two review authors (XYL, TT) independently assessed each included trial for risk of bias according to the following six major criteria, as recommended in the Cochrane Handbook for Systematic Reviews of Interventions: sequence generation, allocation concealment, blinding of patient and personnel, blinding of outcome assessors, incomplete outcome data, and selective outcome reporting. A judgment of either ‘low’, ‘high’ or ‘unclear’ risk with justifications on each criterion was assigned by completing a ‘Risk of bias’ table for each included trial. Any disagreement among the review authors was resolved by discussion leading to a consensus and involved a third review author if necessary.

#### 2.6.2. Treatment Effect for Primary and Secondary Outcomes

Pooled outcome estimates for continuous data were reported using mean difference (MD) if all data were of the same measurement scale, and for categorical data, risk ratios (RRs) were used. We reported the point estimates with their respective 95% confidence intervals (CI). If pooled analyses were not possible due to reasons such as major discrepancies in study characteristics or outcome reporting, as detailed under the assessment of heterogeneity section, we reported the results of the studies individually or in separate subgroups without combining the subgroup estimates.

#### 2.6.3. Missing Data

We followed the recommendations in Section 8.13.2 in the Cochrane Handbook for Systematic Reviews of Intervention in assessing the risk of bias from incomplete outcome data [23].

We performed our analyses for all outcomes, where possible, using intention-to-treat (ITT) data (analysed according to randomisation, irrespective of subsequent discontinuation of the study or deviation from the protocol, if the outcome data of these participants were available or were imputed by the study authors). If there had been missing outcome data that were not imputed, we would have performed a modified ITT analysis (analysed according to randomisation with only available outcome data and without the missing data) [24]. If ITT data were not provided, we included outcome data of the participants either in a ‘per protocol’ or ‘as treated’ manner, as provided by the study authors, but made a corresponding note in the Characteristics of included studies table.

#### 2.6.4. Assessment of Heterogeneity

We used the I2 statistic to quantify the degree of inconsistency in the results [25], with a cut-off of 50% and above considered as the level at which the degree of heterogeneity was of sufficient concern to justify an exploration of possible explanations. In such a situation, we evaluated studies in terms of their clinical and methodological characteristics using the following criteria to determine whether the degree of heterogeneity may be explained by differences in those characteristics, and whether a meta-analysis was appropriate.

We assessed the following criteria:Characteristics of the participants (e.g., age, gender, occupation).Settings of the studies (e.g., community or institution).Interventions (type of chocolate substance given, dosage and length of intervention (dosage: weekly or less frequent vs. twice weekly or more frequent)).Risk of bias, in particular, risks of selection and attrition bias (as detailed in the assessment of risk of bias in included studies section).

If we identified any of the above-mentioned factors during our exploration that we considered to be a plausible explanation of the observed heterogeneity, we separated the studies into subgroups according to the factors concerned if there were sufficient studies in each subgroup.

### 2.7. Reporting Bias

We planned to use a funnel plot to assess any reporting biases where possible, if there were more than 10 included studies for the outcome of interest. If a clear asymmetry had been identified, we would have added a note of caution in our interpretation of the corresponding results. However, as there were no more than 10 included studies identified for any of the outcome of interest, this analysis was not conducted.

### 2.8. Data Synthesis

We pooled the study data and performed meta-analysis if there was more than one study reporting the same outcome and if the included studies were sufficiently homogenous in terms of populations, interventions, comparisons and outcomes measured. We used a random effect model in our meta-analysis using the RevMan 5.4 software [26].

However, if there were marked differences between the study characteristics and reported outcomes, we would have summarised the results of the study narratively.

### 2.9. Subgroup Analysis and Investigation of Heterogeneity

If relevant data had been available, we would have performed subgroup analyses based on the participant characteristics including gender (men, women or other) and age (adults and children); type/form of chocolate given; dosage and length of intervention (dosage: weekly or less frequent vs. twice weekly or more frequent; length of intervention: shorter than 3 months vs. longer).

### 2.10. Sensitivity Analysis

We would have performed sensitivity analysis for primary and secondary outcomes for studies with sufficient numbers available to assess impact of excluding studies with high risk of bias.

### 2.11. Rating Certainty-of-Evidence

We performed certainty-of-evidence rating using the GRADE approach for all the primary outcomes included in this review, and highlighted the major outcomes using one ‘Summary of findings’ table for each comparison. We used the five GRADE criteria (study limitations, consistency of effect, imprecision, indirectness, and publication bias) to assess the certainty-of-evidence for each of these outcomes based on the body of evidence generated by the studies that contributed data to the meta-analyses [27].

Specifically, for the criterion of study limitations, we made the decision on the overall risk of bias across the pool of relevant studies that contributed to each specific outcome rated on two levels: (1) determining the overall risk of bias of any single study, and (2) determining the risk of bias across the pool of relevant studies (namely, overall study limitation). To determine the overall risk of bias of any single study, we assigned the overall risk of bias status of the single study according to the worst risk of bias domain that was relevant to the specific outcome, apart from the domain of selective outcome reporting. To determine the risk of bias across the pool of relevant studies, we referred to the guideline as detailed in Table 12.2.d of the Cochrane Handbook for Systematic Reviews of Intervention [27].

If we identified an issue in any of the five GRADE criteria that we considered to pose a serious enough risk to influence the outcome estimate, we downgraded the certainty of evidence by one level, and when we considered the issue to be very serious, we downgraded the certainty of evidence by two levels [27]. Whenever we decided to downgrade the certainty of evidence from the default high certainty, we justified our decision and described the level of downgrading in the footnotes of the table. We constructed the ‘Summary of findings’ table using an Internet-based version of GRADEpro software [28], according to the methods and recommendations described in Chapter 11 of the Cochrane Handbook for Systematic Reviews of Interventions [25].

## 3. Results

### 3.1. Description of Studies

#### 3.1.1. Results of the Search

The initial search through MEDLINE (PubMed) and CENTRAL (which covered PubMed, EMBASE, CINAHL, and trial registry (Clinicaltrial.gov and ICTRP)) databases identified 2155 records with 1456 records remaining after removing duplicates. Of these, 101 articles appeared to be relevant after we inspected the titles. We further evaluated the 101 articles by reading the abstracts and/or full-texts, excluding 69 records in the process, with 17 studies that appeared to be on-going in the form of trial registry records. Finally, we included 18 records in our analysis. Four out of the eighteen records reported different outcomes for the same study [29], and they were merged into a single study, leaving fifteen distinct studies included in our analysis. The flow diagram of the studies from the initial search to meta-analysis is shown in Figure 1. We describe all the included studies in the ‘Characteristics of included studies’ table. The reasons for exclusion after inspection of their full-texts mostly consist of study subjects having co-morbidities (n = 22), not relevant in terms of study, outcome or intervention (n = 23), on-going trials (n = 17), questionnaire-based studies (n = 6), review papers (n = 6), case report or single group study (n = 2), comparison similar to intervention (n = 5), repeated or duplicated study (n = 3), and no outcome of interest (n = 2). To enable ease of reading, chocolate and cocoa products will be collectively referred to as chocolate from this point onwards.

#### 3.1.2. Included Studies

Among the 15 included studies, 9 were parallel-group, individually-randomised, two-arm RCTs. One study [30] was a five-arm controlled trial with equal sample size, one study was a RCT trial involving two phases [31] and another four studies [32,33,34,35] were crossover trials. Fifteen studies were conducted in eight different countries, namely United States of America (USA) [33,34,36,37], Japan [35,38,39], Spain [29,40,41], China [32], Korea [42], Russia [30], Australia [31] and Portugal [43].

Ten studies included participants of both genders [30,31,32,33,34,35,37,38,41,43]. The remaining studies include participants of either male or female gender [29,36,39,40,42]. The age of the participants ranged from around 20 years [38] to 69 years [31,37]. To stay true to our objective of reviewing the health effects of chocolate and cocoa product consumption in the general healthy population, all participants recruited into the studies were either healthy individuals [30,31,32,34,36,37,38,39,40,41,42,43] or had an established clinical diagnosis of acne [33], post-menopause [29] or were pre-diabetic [35], without diagnosis of any other health comorbidities. Participants were recruited from a combination of clinical and non-clinical settings. Four studies recruited participants that were from universities or colleges [32,33,38,43], three studies were from healthcare facilities [29,30,42], four studies were of volunteers [31,34,37,41], and the remaining three studies did not mention the source of recruitment [35,39,40]. The total sample size for the chocolate intervention group was 525 participants compared to 500 participants for placebo. Duration of the eight studies ranged from four to twenty-four weeks.

The interventions investigated were in various forms, including chocolate bars [31,33,37,38], beverage or snacks [34,36,39,40,42], pharmaceuticals (capsules and cocoa extract) [30,35,41], and others not clearly mentioned [29,32,43]. In terms of comparison, eight studies compared with placebo [31,33,35,37,40,41,42,43], another five studies with other interventions [30,32,34,36,39], while the remaining two studies compared with no interventions [29,38].

The characteristics of the included studies are shown in Table 1.

#### 3.1.3. Outcomes

Out of the fifteen studies, a total of 10 studies reported anthropometric measurements [29,31,32,34,36,38,39,40,41,43] while the remaining four studies reported on the more focused health parameters of acne [33], gut microbiome [30], skin condition [42], neuropsychological functioning [37], and diabetes [35].

### 3.2. Risk of Bias Assessment

The proportion of studies with low, high, and unclear risk of bias in each domain is illustrated in Figure 2. Overall, there was a wide variation in the risk of bias of the studies across six domains, with the majority (50% and above) of the included studies judged to have low risk of bias in the domains of random sequence generation, incomplete outcome data and selective reporting. Only a minority (<25%) have low risk of bias in blinding of outcome assessment, allocation concealment, as well as blinding of participants and personnel. On the other hand, a small but significant proportion of studies (25% and above) were judged to have high risk of bias in the domains of blinding of participants and personnel, selective reporting and other bias. Meanwhile, a large proportion (50% and above) of included studies did not provide sufficient information to enable meaningful assessment on the risks of bias in the domains of random sequence generation, allocation concealment, blinding of participants, and personnel and blinding of outcome assessment.

### 3.3. Effects of Intervention

In total, 15 studies with 1025 participants contributed to meta-analyses of the data, while the outcome data of three studies [33,35,42] were not reported sufficiently for meta-analysis. Thirteen major outcomes were evaluated, namely systolic blood pressure [29,32,34,36,37,38,39,41,43], diastolic blood pressure [29,32,34,36,37,38,39,41,43], total cholesterol [32,34,36,37,39], triglyceride [30,32,34,36,37,39], low density lipoprotein (LDL) [30,32,34,36,37,39,41], high density lipoprotein (HDL) [30,32,34,36,37,39], body weight [29,34,36,38,41], BMI [29,31,32,34,36,38,39,40,43], waist circumference [34,36,38,41], body fat percentage [29,36,38,40,41], fasting plasma glucose [32,34,36,38,41], cognitive function (attention) [29,37] and cognitive function (processing speed and cognitive flexibility) [29,37].

#### 3.3.1. Skin Condition

Two studies [30,42] measured different skin parameters which did not show any significant difference in all the skin outcomes reported, including skin hydration, wrinkle severity, sebum droplet size, corneocyte damage, corneocyte exfoliation rate and skin elasticity. The graphical representations of all the above findings are shown in Appendix A.

#### 3.3.2. Blood Pressure

Results show that consuming chocolate (369 participants) was not significantly better than consuming control (394 participants) in reducing systolic blood pressure (MD = −0.20 mmHg; 95% CI = −1.70, 1.29; nine studies; I^2^ = 40%) and diastolic blood pressure (MD = −0.05 mmHg; 95% CI = −1.13, 1.03; nine studies; I^2^ = 31%). Certainty-of-evidence was moderate for the three trials due to unclear or high risk of bias for allocation concealment, blinding of participants and personnel, and blinding of outcome assessor. The graphical representation of these findings is shown in Figure 3 and Figure 4.

#### 3.3.3. Lipid Profile

Results show that chocolate consumption (229 participants) did not produce significantly different effects compared to control (258 participants) on total cholesterol (MD = 3.59 mg/dL; 95% CI = −0.14, 7.31; five studies; I^2^ = 0%). All five studies where considerably homogenous with I2 of 0%. Certainty-of-evidence was moderate for the five trials due to unclear bias or high risk of bias for random sequence generation, allocation concealment, blinding of participants and personnel, and blinding of outcome assessor [32,34,36,37,39]. The graphical representations of these findings are shown in Figure 5.

Results show that chocolate consumption (235 participants) significantly reduces triglyceride levels compared to control (276 participants) (MD = −3.86 mg/dL; 95% CI = −7.72, 0.00; six studies; I^2^ = 31%). Certainty-of-evidence was moderate for the six trials due to unclear bias or high risk of bias for random sequence generation, allocation concealment, blinding of participants and personnel and blinding of outcome assessor. There was a high risk of bias among papers for other bias [30,32,34,36,37,39]. The graphical representations of these findings are shown in Figure 6.

Results show that chocolate consumption (258 participants) did not produce significantly different effects compared to control (300 participants) on LDL (MD = 0.79 mg/dL; 95% CI = −2.17, 3.74; seven studies; I^2^ = 43%). Certainty-of-evidence was moderate for the seven trials due to unclear bias or high risk of bias for random sequence generation, allocation concealment, blinding of participants and personnel, and blinding of outcome assessor. There was a high risk of bias among papers for other bias which are mainly due to industrial sponsored trials and risk of carry over effect in cross-over trials [30,32,34,36,37,39,41]. The graphical representations of these findings are shown in Figure 7.

Results show that chocolate consumption (235 participants) did not produce significantly different effects compared to control (276 participants) on HDL (MD = 0.18 mg/dL; 95% CI = −1.00, 1.36; six studies; I^2^ = 71%). Certainty-of-evidence was low for the six trials due to unclear bias or high risk of bias for random sequence generation, allocation concealment, blinding of participants and personnel, and blinding of outcome assessor and substantial heterogeneity. The graphical representations of these findings are shown in Figure 8.

#### 3.3.4. Anthropometric Parameters

Results show that consuming chocolate (210 participants) was not significantly different when compared to consuming control (168 participants) in changing body weights (MD = −2.40 kg; 95% CI = −7.27, 2.47; five studies; I^2^ = 94%). The certainty-of-evidence was considered very low for the five trials due to high risk of bias for blinding of participants and personnel, and other bias, substantial heterogeneity and imprecision [29,34,36,38,41]. The graphical representation of these findings is shown in Figure 9.

Results show that consuming chocolate (323 participants) was not significantly different from consuming control (364 participants) in affecting body mass index (MD = −0.09 kg/m^2^; 95% CI = −0.24, 0.07; nine studies; I^2^ = 0%). Certainty-of-evidence was considered moderate for the nine trials due to unclear or high risk of bias for allocation concealment, blinding of participants and personnel, and blinding of outcome assessor. The graphical representation of these findings is shown in Figure 10. All nine studies where considerably homogenous with I^2^ of 0% [29,31,32,34,36,38,39,40,43].

Results show that consuming chocolate (143 participants) was not significantly better than consuming control (107 participants) in reducing waist circumference (MD = −0.33 cm, 95% CI = −1.71, 1.04; four studies; I^2^ = 0%). Certainty-of-evidence was moderate for the four trials due to unclear or high risk of bias for allocation concealment, blinding of participants and personnel, blinding of outcome assessor, and other bias. The graphical representation of these findings is shown in Figure 11. All four studies where considerably homogenous with I2 of 0% [34,36,38,41].

Results show that consuming chocolate (151 participants) was not significantly better than consuming control (148 participants) in reducing body fat percentage (MD = −0.58%; 95% CI = −1.62, 0.47; five studies; I^2^ = 3%). Certainty-of-evidence was moderate for the five trials due to unclear or high risk of bias for allocation concealment, and blinding of participants and personnel [29,38,40]. The graphical representation of these findings is shown in Figure 12.

One study [32] measured additional anthropometric parameters individually and did not show any significant difference in all the outcomes measured, which is waist hip ratio. The graphical representations of all above findings are shown in Appendix A.

#### 3.3.5. Blood Glucose

Results show that consuming chocolate (210 participants) was not significantly different from control (241 participants) in affecting fasting plasma glucose (MD = 1.14 mg/dL; 95% CI = −0.50, 2.77; five studies; I^2^ = 1%). Certainty-of-evidence was moderate for the five trials due to unclear or high risk of bias for allocation concealment, blinding of participants and personnel, blinding of outcome assessment, and other bias. The graphical representation of these findings is shown in Figure 13. All four studies where considerably homogenous with I^2^ of 1% [32,34,36,38,41].

#### 3.3.6. Cognitive Function

Results show that consuming chocolate (116 participants) was not significantly different from control (111 participants) in affecting attention time for completing tasks (MD = −0.67 s; 95% CI = −3.38, 2.05; two studies; I2 = 0%) and processing speed and cognitive flexibility (MD = −3.14 s; 95% CI = −11.55, 5.28; two studies; I2 = 67%). Certainty-of-evidence was low for the two trials due to unclear or high risk of bias for most of the domains and heterogeneity [29,37]. The graphical representation of these findings is shown in Figure 14 and Figure 15.

Several other different cognitive function parameters were also reported by the same two studies [29,37]. However, all of these outcomes were distinctively different from each other (i.e., not suitable to be pooled) and, therefore, did not show any significant differences in all the outcomes measured. There were an additional five cognitive function tests: (a) selective reminding test (nine parameters: immediate free recall, long-term storage, short-term recall, long-term retrieval, consistent long-term retrieval, random long-term retrieval, cued recall, delayed free recall and delayed recognition), (b) Ctroop Colour and Word test (three parameters: word, colour and colour-word), (c) Wechsler Memory Scale-III (two parameters: Faces I, Faces II), (d) Wechsler Adult Intelligence Scale-III (digit symbol), and (e) Activation-Deactivation Adjective Check List (general activation subscale) [37], with additional cognitive function parameters on working memory [29], and, therefore, did not show any significant differences in all the outcomes. The graphical representations of all above findings are shown in Appendix A.

#### 3.3.7. Quality of Life

Only one study [29] measured quality of life and, therefore, did not show any significant differences in all the outcomes measured, which consist of five dimensions evaluated in the EQ-5D-3L (mobility, self-care, usual activities, anxiety/depression and pain/discomfort). The graphical representations of all above findings are shown in Appendix A.

### 3.4. Safety Assessment

Safety assessment (as analysed based on withdrawals and adverse effects across trials) is descriptively summarised in Table 2. Six studies did not provide any information on reasons for withdrawals or adverse effects [32,35,36,40,41,43]. Three studies [29,30,33] addressed withdrawals but did not specifically report the safety assessment throughout the trials. In these three articles, one dropout case was reported due to gastrointestinal disturbances caused by chocolate [33].

The remaining six studies reported on safety assessment. Two studies reported that cocoa flavanol supplementation or cocoa powder is well tolerated without any subjective adverse events. No significant changes in serum biochemistry, haematologic indices, and urinalysis (plasma total protein, albumin, uric acid, free fatty acids, phospholipids, total bilirubin, aspartate aminotransferase (AST), alanine transaminase (ALT), glucose, blood urea nitrogen, creatinine, gamma-glutamyltranspeptidase (GGT), alkaline phosphatase (ALP), lactate dehydrogenase, sodium, potassium, chloride, proteinuria, glucosuria, urobilinogen, occult blood, and hemoglobin concentrations and hematocrits) were reported across the same two studies [39,42]. One study reported a slight increase in resting glucose levels, defined as blood glucose levels at rest, and before the end of six minutes of exercise during the study (especially in the intervention group with normal diet +20 g/day of high-cocoa chocolate) [38], though another study suggested that cocoa products (sweetened and unsweetened) did not adversely affect body weight during short term consumption [34]. Two dropouts due to unpalatability of dark chocolate were also reported [31]. A total of 13 adverse events were reported in the treatment group compared to 10 in the control group. Most were mild to moderate including gastrointestinal disturbances and cold symptoms. One severe adverse event of atrial arrhythmia (type unknown) was reported in the treatment group and the participant was hospitalised. This event was thought to be not related to the treatment [37].

## 4. Discussion

### 4.1. Summary of Main Findings

In this meta-analysis of RCT studies, the only significant finding identified was that chocolate consumption significantly reduced triglycerides compared to control, while the effects of chocolate on all other outcome parameters were not significant from the control group.

The findings relating to no significant changes in blood pressure, lipid profile (except for triglyceride), anthropometric parameters, and blood glucose, which are possible contributors of cardiovascular risk (CVD), contradict the findings from a systematic review conducted in 2006 which suggested that flavonoids from chocolate are likely protective against coronary heart disease mortality [44]. At that point, the authors (Ding et al.) recommended the need for a meta-analysis to confirm their report and highlighted the need to conduct long-term randomised feeding trials to ensure a more accurate outcome assessment beyond short-term studies of CVD risk factor intermediates. Another systematic review that included 14 prospective cohort studies and two cross-sectional studies covering 344,453 participants of both genders showed that consuming chocolate had beneficial effects in reducing CVD risk, especially among women. However, the limitation of this study was that assessment of the differences in and contributions of test items (e.g., type of chocolate and percentage of cocoa content) was not considered [45]. A meta-analysis which included 42 acute and short-term chronic (less than 18 weeks) RCTs involving 1297 participants with CVD risk but not critically ill showed promising effects on biomarkers of CVD risk. Many of the included studies in this review were industrial-sponsored and of short duration [46]. Similar limitations of short duration of studies and insufficient details on intervention are also supported by another systematic review [47]. A meta-analysis to investigate the association between chocolate and coronary artery disease (CAD) which included six prospective studies with a median follow up period of 8.78 years and 336,289 participants concluded that consumption of chocolate for more than one time per week or more than 3.5 times per month reduced risk of CAD, though its analysis was limited by dietary and lifestyle confounders not being considered [48]. A different meta-analysis conducted, which focused more on the association of chocolate with cardiometabolic disorders covering seven studies (one cross sectional study and six cohort studies), with 114,009 participants showed that higher levels of chocolate consumption may be associated with a one-third reduction in the risk of CVD. However, the included studies were mostly conducted in Europe and USA and do not represent other countries’ populations [49]. Compared with these systematic reviews and meta-analyses, our review focused on a healthy population, which has not been assessed as a single study population on the global health effects of chocolate. To improve the robustness of our analysis on the meaningful contribution of longer consumption duration of chocolate in producing changes in bodily functions, we only included studies that investigated chocolate consumption for four weeks or longer. During our screening we excluded studies that investigated the effects of acute ingestion of chocolate on cardiovascular risk factors such as blood pressure, which in our opinion may not be a good representation of its overall global effects. We have also limited our inclusion criteria to randomised controlled trials. Similar to past reviews, we observed high heterogeneity in the intervention details of chocolate or cocoa products, with limited quantitative analysis of chemical or biomarkers.

Interestingly, although there were no significant differences in pooled effects of either control or chocolate groups on any outcome parameters, through visual comparison of the individual forest plots of all cardiovascular related outcomes (anthropometric parameters, glucose, lipid profile, blood pressure), one study [30] showed high preference for the comparator group having a better HDL profile than the chocolate/cocoa group. This observation was maybe contributed to by the interventions used in the control group (which was not a placebo or non-treated group), which consisted of an additional lycopene-containing formulation of medium and polyunsaturated fatty acids. This study conducted by Wiese (2019) primarily focused on the effects of chocolate and lycopene on gut microbiome, while investigating their effects on lipid profiles as additional outcomes. Lycopene is a potent antioxidant belonging to a group of compounds called carotenoids, which are primarily found in vegetables and fruits [50,51]. A recent systematic review reported significant beneficial effects of lycopene-rich formulations on metabolic parameters of lipid profiles and blood glucose level regardless of duration and dose, based on three interventional studies. However, no meta-analysis was conducted to pool these outcomes [52]. On the other hand, the same study by Wiese showed that the chocolate group had significantly lower triglycerides compared to its comparators of lycopene-containing interventions. Though the reasons remain unclear, this difference can be due to the effect of the contribution of medium saturated fatty acids given in combination with lycopene [30]. The comparison of lycopene as an individual compound with varying flavanol concentrations in chocolate may be further investigated to provide further understanding to such findings.

Our findings of no differences in the effect of chocolate or cocoa products on anthropometric parameters were similar with the findings from a meta-analysis of 35 RCTs conducted by Kord-Varkaneh, which reported that cocoa or dark chocolate supplementation did not affect body weight, BMI, and waist circumference in comparison with control. Further subgroup analyses from Kord-Varkaneh (2018) indicated that weight and BMI were reduced with cocoa or dark chocolate supplementation at doses of more than 30 g chocolate per day for a period of four to eight weeks trial [53]. In our review of only RCTs, the highest dose of dark chocolate used in studies that reported anthropometric parameters was 10 g/day (99% cocoa) which amounts to 59 kcal, and did not increase body weight [29].

Our findings relating to cognitive function parameters contradict a systematic review by Scholey (2013). In their review, the authors did not perform meta-analysis to quantitatively pool the results, although three out of the seven included studies showed positive cognitive function effects. It was suggested that further investigation was needed of acute cognitive effects of chocolate along with examination of functional brain changes associated with cocoa-flavanols [54]. Recent human trials have shown that dark chocolate (35 g) improved verbal memory in health young adults, which further supports exploration in this area [12]. In addition to their effects on memory, functional and all rounded assessments utilising physiological and cognitive challenge tasks are also important factors to consider when evaluating the effects of cocoa-flavanol intake on brain function [55].

As for the controversial yet popular topic regarding the effects of chocolate on skin parameters, our findings were supported by other studies which found no statistically protective or adverse effects of high flavanol chocolate on skin sensitivity to ultraviolet radiation based on a 12-week RCT with 74 women participants [56]. An earlier RCT of 30 healthy subjects conducted in 2009 suggested that high flavanol chocolate can be photoprotective, but this study has methodological limitations whereby patient characteristics and between-group comparisons were not reported. Although two human trials have shown that consuming 10 g to 25 g of dark chocolate daily for four weeks can exacerbate acne by enhancing corneocyte shedding in the outer layers of the skin and promoting bacterial colonisation of the residual skin surface components [16,57], there are major limitations to these studies which include the absence of a control group. From our review, there were limited RCTs that specifically address the effect of chocolate on acne. Therefore, future randomised controlled trials are needed to better understand the possible adverse effects of chocolate consumption on acne specifically.

### 4.2. Assessment of the Overall Certainty-of-Evidence (GRADE Approach)

Overall, the certainty-of-evidence for most of the outcomes ranged from low to moderate, with the main concerns being: study limitations, as illustrated by a combination of high or unclear risk of bias in one or more domains of the included studies; inconsistency, as shown by the substantial heterogeneity without plausible explanation; and imprecision, as indicated by the wide range of effects in the 95% confidence intervals. The certainty-of-evidence for the outcome of body weight was rated as very low due to major concerns in all three areas mentioned.

The graphical representations of all above findings are shown in Appendix A.

### 4.3. Limitation and Strength of This Study

These findings are limited by the substantial degree of heterogeneity between trials with no plausible explanation, which could not be explained by pre-specified subgroup analyses, including blinding, flavanol content of the control groups, age of participants, or study duration. Due to limited quantitative data being available on chemical or biomarkers of chocolate/cocoa product, we were also unable to assess the possible contribution of specific bioactive constituents of chocolate. The duration of the 15 RCTs conducted are mostly short-term between four weeks (six studies) and six weeks (two studies).

Our predefined criteria with regards to intervention have resulted in some possibly useful studies to be excluded. For example, we excluded Grassi et al. (2015) [58] on the basis of duration of intervention, as the study only evaluated the use of chocolate for one week, which was short of our pre-specified duration of four weeks. We note that the findings of the study were relied on in the recommendations of the European Food Safety Authority. While we believe our pre-defined inclusion criteria were justifiable, we acknowledge the possibility of having excluded some potentially useful evidence.

The strength of this review is the comprehensive literature search including several databases, trial registries, and reference lists of included trials. To the best of our knowledge, this is the first review to focus mainly on healthy human subjects, namely, those without a diagnosis of any medical condition as participants. We successfully covered a majority of the relevant and popular health parameters of chocolate which are markers for skin condition, CVD, diabetes, and cognitive function. Our assessment of data was focused on the highest level of evidence, which are RCTs. Furthermore, we limit the articles’ inclusion criteria to studies that reported the effects of a chocolate-containing intervention (intervention group) to a comparator group which did not consume chocolate or cocoa products in any form. All these factors helped in reducing the effects of known confounders reported in previous systematic reviews and meta-analyses. In order to obtain more accurate data and minimise statistical errors, attempts were made to contact some of the individual study authors for clarification of data.

## 5. Conclusions

Low-to-moderate evidence showed no clear differences between the effects of chocolate and control groups on parameters related to skin, blood pressure, lipid profile, cognitive function, anthropometry, blood glucose, cognitive function, and quality of life regardless of form, dose, and duration among healthy individuals. Chocolate is generally well accepted by study subjects, with gastrointestinal disturbances and unpalatability being the most reported concerns. There is a wide variation in terms of form of chocolate or cocoa products investigated, limited data on quantitative analysis of any biomarkers or chemical markers and studies conducted are short-term in duration. High-quality evidence on the influence of chocolate on skin-related adverse effects, especially acne, cognitive functions, and quality of life, are still limited. There were no RCTs identified for anticancer and immune-enhancing effects of chocolate among healthy volunteers.

## Figures and Tables

**Figure 1 nutrients-13-02909-f001:**
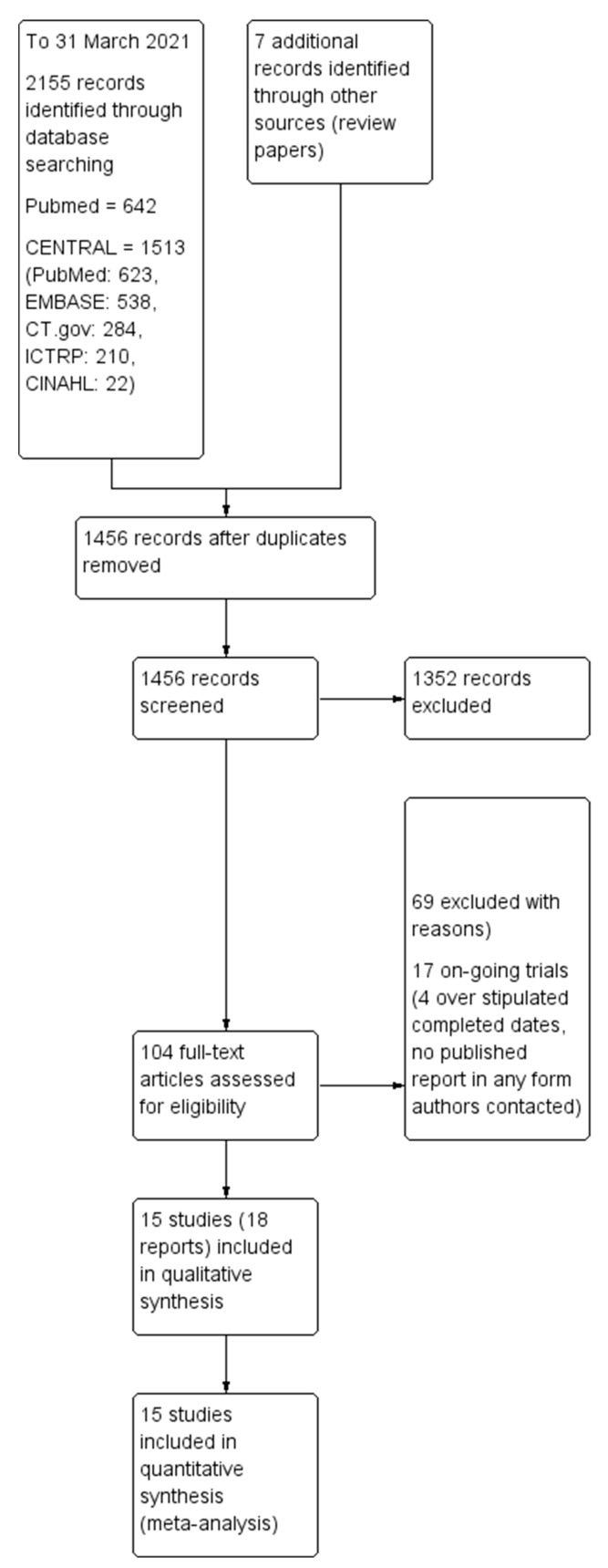
Preferred reporting items for systematic reviews and meta-analyses (PRISMA) flowchart that shows study flow in the review work to investigate the benefits of chocolate.

**Figure 2 nutrients-13-02909-f002:**
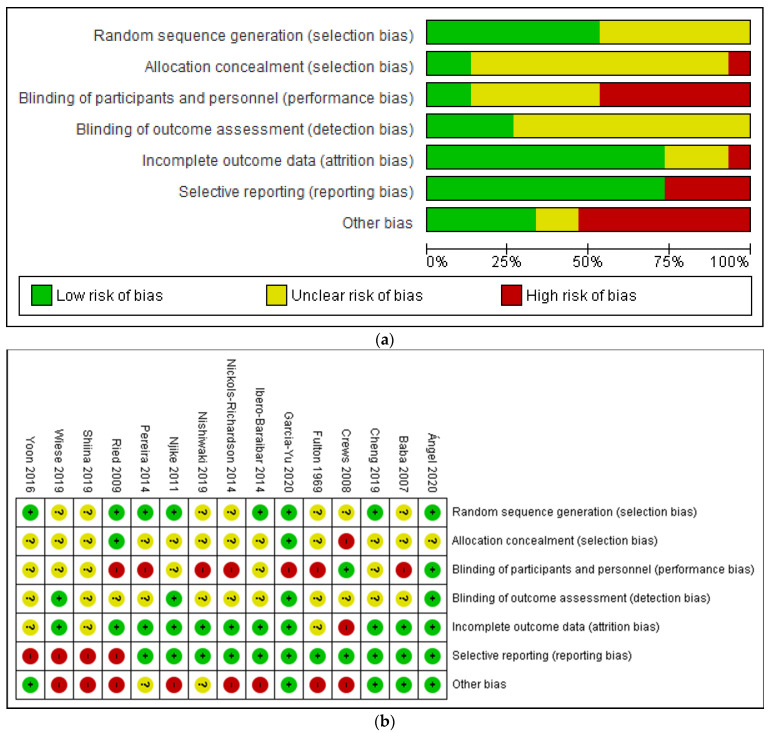
(**a**) Risk of bias assessment (ROB) graph and (**b**) ROB summary of included studies based on authors’ judgment.

**Figure 3 nutrients-13-02909-f003:**
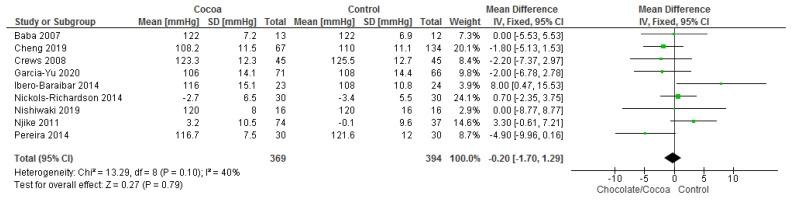
Forest plot of comparison: chocolate versus control, with the outcome-systolic blood pressure (mmHg).

**Figure 4 nutrients-13-02909-f004:**
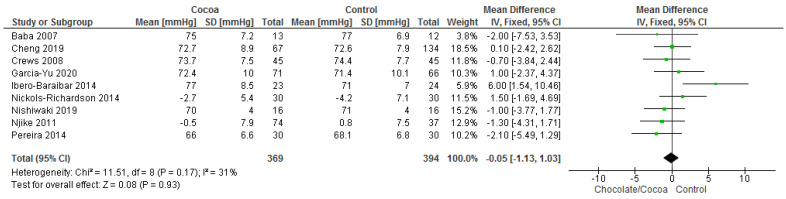
Forest plot of comparison: chocolate versus control, with the outcome-diastolic blood pressure (mmHg).

**Figure 5 nutrients-13-02909-f005:**
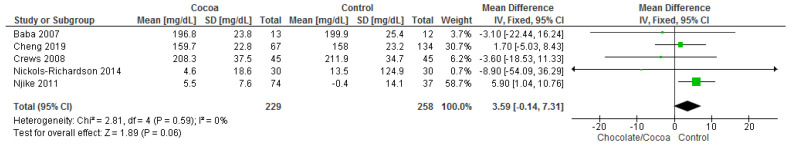
Forest plot of comparison: chocolate versus control, with the outcome-total cholesterol (mg/dL).

**Figure 6 nutrients-13-02909-f006:**
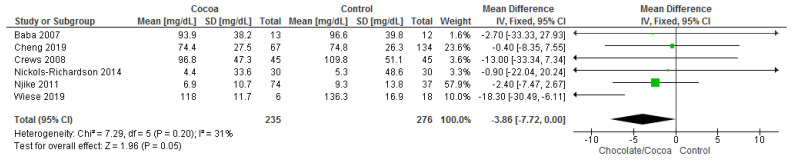
Forest plot of comparison: chocolate versus control, with the outcome-triglyceride (mg/dL).

**Figure 7 nutrients-13-02909-f007:**
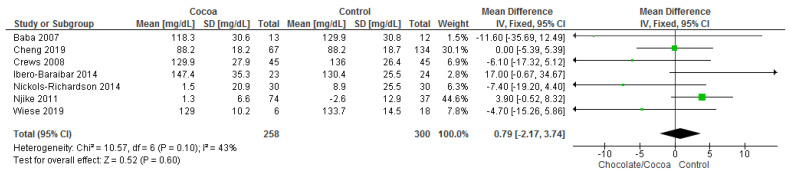
Forest plot of comparison: chocolate versus control, with the outcome-low density lipoprotein (mg/dL). Nichols-Richardson (2014) and Nijike (2011) reported mean changes and their respective SD, instead of mean scores.

**Figure 8 nutrients-13-02909-f008:**
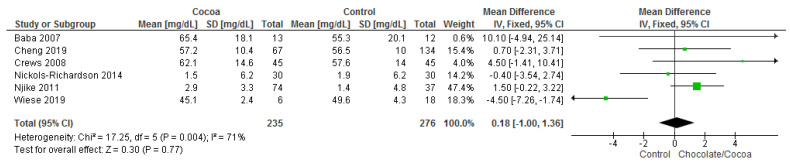
Forest plot of comparison: chocolate versus control, with the outcome-high density lipoprotein (mg/dL). Nichols-Richardson (2014) and Nijike (2011) reported mean changes and their respective SD, instead of mean scores.

**Figure 9 nutrients-13-02909-f009:**
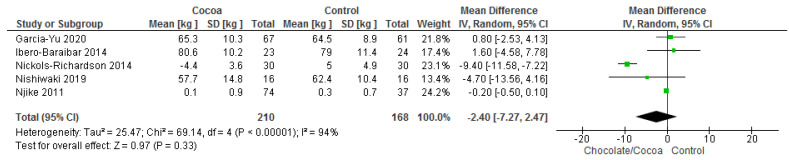
Forest plot of comparison: chocolate versus control, with the outcome-body weight (kg). Nichols-Richardson (2014) and Nijike (2011) reported mean changes and their respective SD, instead of mean scores.

**Figure 10 nutrients-13-02909-f010:**
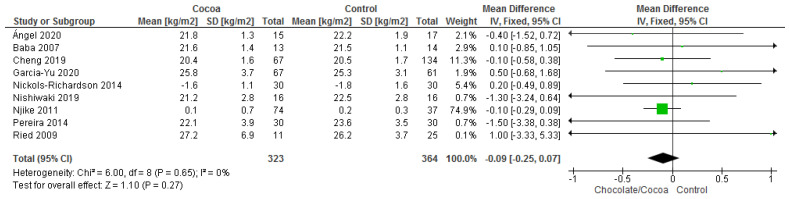
Forest plot of comparison: chocolate versus control, with the outcome—body mass index (kg/m^2^). Nichols-Richardson (2014) and Nijike (2011) reported mean changes and their respective SD, instead of mean scores.

**Figure 11 nutrients-13-02909-f011:**
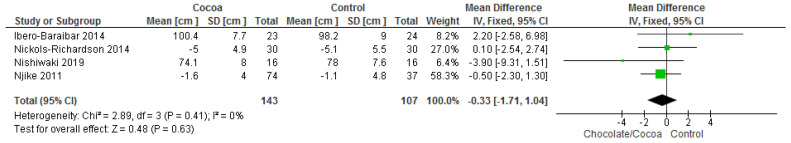
Forest plot of comparison: chocolate versus control, with the outcome-waist circumference (cm).

**Figure 12 nutrients-13-02909-f012:**
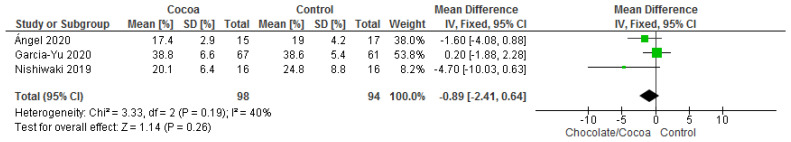
Forest plot of comparison: chocolate versus control, with the outcome-body fat percentage (%).

**Figure 13 nutrients-13-02909-f013:**
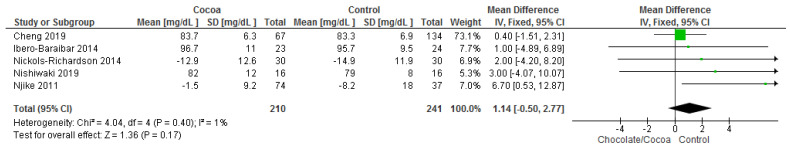
Forest plot of comparison: chocolate versus control, with the outcome—fasting plasma glucose (mg/dL). Nichols-Richardson (2014) and Nijike (2011) reported mean changes and their respective SD, instead of mean scores.

**Figure 14 nutrients-13-02909-f014:**
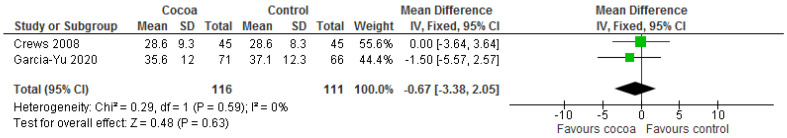
Forest plot of comparison: chocolate versus control, with the outcome-trail marking test (attention) (seconds).

**Figure 15 nutrients-13-02909-f015:**
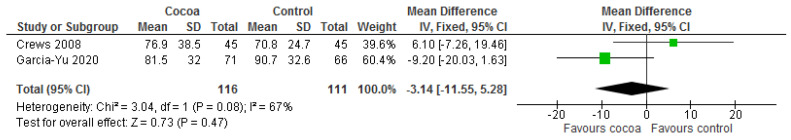
Forest plot of comparison: chocolate versus control, with the outcome—trail marking test (processing speed and cognitive flexibility) (seconds).

**Table 1 nutrients-13-02909-t001:** Characteristics of included studies.

Author, Country, Year	Clinical Trial Design	Population	Sex	Sample Size Chocolate/Placebo	Duration/Outcome	Intervention Group	
						Intervention Group	Placebo Group
Ángel García-Merino, Spain, 2020	Randomised, parallel-group placebo-controlled trial	Male endurance cross-country athletes	Male	15/17	10 weeks	5 g of fat-reduced cocoa containing 425 mg of flavanols	5 g of maltodextrin
Fulton, USA, 1969	Crossover, single-blind	Subjects with mild to moderate acne	Both	65	2 months	114 g of bittersweet chocolate bar	112 g of 28% vegetable fat to mimic the lipids contained in chocolate liquor and cocoa butter bar
Cheng, China, 2018	Randomised crossover 33 Latin-square design	Male or female aged 20–40 years	Both	67	4 weeks	Cocoa butter	(1) Palm olein (2) extra virgin olive oil
Garcia-Yu, Spain, 2020	Controlled randomised trial with two parallel groups	Women aged between 50 and 64 years and in the period of post-menopause	Female	69/63	6 months	Chocolate (99% cocoa) 10 g as supplement	No intervention
Nishiwaki, Japan, 2019	Randomised, controlled, parallel-group intervention study	Healthy Japanese college student	Both	16/16	4 weeks	20 g/day (508 mg of cacao polyphenol) of high-cocoa chocolate	No intervention
Wiese, Russia, 2019	Randomised, parallel-five group placebo-controlled trial	Moderately obese volunteers	Both	6 people per group	4 weeks	10 g of dark chocolate	(1) 7 mg GA lycopene (GAL) formulated with medium saturated fatty acids (GAL-MUFA) (2) 30 mg GAL-MUFA (3) 30 mg GAL-PUFA
Yoon, Korea, 2015	Randomised, parallel-group placebo-controlled trial	Healthy female volunteers	Female	31/31	24 weeks	Beverage containing 4 g cocoa powder to yield 320 mg total cocoa flavanols	Nutrient-matched cocoa-flavored beverage without cocoa flavanols
Shiina, Japan, 2019	Randomised, placebo-controlled doubleblind crossover trial	Pre-diabetic volunteers	Both	11/11	4 weeks	Cacao procyanidin supplement (83.3 ± 2.7 mg/day) which contain 13.9 ± 2.7 mg procyanidins	240 mg dextrin
Baba, Japan, 2007	Randomised controlled trial	Healthy Japanese male subjects	Male	25	12 weeks	Cocoa powder and sugar	Sugar
Ibero-Baraibar, Spain, 2014	Randomised, parallel and double-blind study	Healthy Caucasian adult	Both	50	4 weeks	Cocoa extract	Placebo
Nickols-Richardson, USA, 2014	Randomised controlled trial	Overweight otherwise healthy women age 25–45 years (premenopausal)	Female	60	18 weeks	Cocoa beverage with dark chocolate	Cocoa free beverage with non- chocolate snacks
Njike, USA, 2011	Randomised, controlled, crossover trial	Overweight, but otherwise healthy, men and women	Both	44	6 weeks	Unsweetened or sweetened cocoa beverage	Non-cocoa beverage
Prereira, Portugal, 2014	Randomised controlled trial	Clinically healthy individuals of Portuguese nationality, all undergraduate students at the Superior Polytechnic Institute of Coimbra, under the age of 25 years	Both	60	4 weeks	Dark chocolate	Placebo
Ried, Australia, 2009	Randomised controlled trial (2 phases)	Prehypertensive otherwise healthy adults	Both	36	8 weeks	Dark chocolate bar	(1) Placebo (2) Tomato extract
Crews, USA, 2008	Double-blind, placebo-controlled, randomised trial	Healthy older male and female adults 60 years and above	Both	101	6 weeks	Dark chocolate bar	Placebo

**Table 2 nutrients-13-02909-t002:** Safety assessment of included studies.

Author, Year	Findings
Angel García-Merino, 2020	Not reported
Fulton, 1969	Not specifically reported; but caused gastrointestinal disturbances in one case leading to defaulting intervention
Cheng, 2018	Not reported
Garcia-Yu 2020	Not specifically reported; but did not change body composition
Nishiwaki, 2019	Slight increase in resting glucose levels (especially in the intervention group with normal diets +20 g/day of high-cocoa chocolate)
Wiese, 2019	Not specifically reported; but did not cause significant changes in glucose and liver enzymes AST and ALT
Yoon, 2015	Well tolerated, no subjective adverse events reported. No significant changes in serum biochemistry and haematologic indices (AST, ALT, glucose, blood urea nitrogen, creatinine, hemoglobin, hematocrits)
Shiina, 2019	Not reported
Baba, 2007	All biochemical and urinalysis within normal range at baseline and at 12 weeks (including plasma total protein, albumin, glucose, uric acid, urea nitrogen, creatinine, free fatty acids, phospholipids, total bilirubin, AST, ALT, GGT, alkaline phosphatase, lactate dehydrogenase, sodium, potassium, chloride, proteinuria, glucosuria, urobilinogen, and occult blood).
Ibero-Baraibar, 2014	Not reported
Nickols-Richardson, 2014	Not reported
Njike, 2011	Cocoa products (sweetened and unsweetened) does not adversely affect body weight during short term consumption
Prereira, 2014	Not reported
Ried, 2009	Dark chocolate: unpalatable (n = 2)-withdrew;Tomato extract: gastrointestinal upset (n = 1)-withdrew
Crews, 2008	13 adverse events reported in treatment group compared to 10 in control group. Most are mild to moderate including gastrointestinal disturbances and cold symptoms. One severe adverse event atrial arrythmia (type unknown) was reported in the treatment group and was hospitalised. This event was thought to be not related to the treatment

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
