# Peer review of "The Health Effects of Chocolate and Cocoa: A Systematic Review"

_nutrients, 2021, doi:10.3390/nu13092909_

Round 1

Reviewer 1 Report

The article is well organized. I only have a few comments.

1) Lines 30-34. Too much simplification in the description of the procedure for obtaining chocolate causes some distortions. First, cocoa beans can be roasted as whole beans, cocoa nibs or cocoa liquor. The description suggests that the Authors mean roasting the beanns. Then, specify the individual stages of chocolate production in the following sequence: cleaning of beans, roasting, dehusking, grinding into nibs, refining (optional), grinding into cocoa liquor, refining (optional), mixing chocolate ingredients (ingredients depending on the type of chocolate: dark, milk, white), conching, tempering, forming, packing. 

Please add two references in line 34: 

- Di Mattia, C.D., Sacchetti, G., Mastrocola, D. and Serafini, M. From cocoa to chocolate: the impact of processing on in vitro antioxidant activity and the effects of chocolate on antioxidant markers in vivo. Frontiers in Immunology, 2017. 8(Article 1207): 1–7.
- Żyżelewicz, D., Budryn, G., Oracz, J., Antolak, H., Kręgiel, D. and Kaczmarska, M. The effect on bioactive components and characteristics of chocolate by
functionalization with raw cocoa beans. Food Research International, 2018. 113: 234–244.

2) Line 35: Theobroma cacao write in italics.

3) Line 36: "... and is rich is polyphenols ...". It should be: "... and is rich in polyphenols ...".

4) Line 38: Add the reference: Å»yżelewicz, D., Krysiak, W., Oracz, J., Sosnowska, D., Budryn, G. and Nebesny, E. The influence of the roasting process conditions on the polyphenol content in cocoa beans, nibs and chocolates. Food Research International, 2016. 89: 918–929.

5) Lines 45-49: You can't write the sentence like this "The 45 cognitive benefits of chocolate are further supported by a recent systematic review which 46 reported improvement in cognitive score or tasks performance among young adults (less 47 than 25 years old) and children with chronic chocolate consumption, again possibly at-48 tributed to flavanols or polyphenols.". Substantive error. Flavanols are a group of polyphenols, so you can't write "flavanols or polyphenols". Rather, it should have been written "... attributed to polyphenols, including flavanols.".

6) Please put the full stops after title of figure 1 and tables 1 and 2.

Author Response

Greetings to Reviewer 1, thanks for your comments. Please refer to the attachment below.

Reviewer 2 Report

The article deals with a very interesting issue, because chocolate in the opinion of consumers is a product not recommended in the diet. The presented analysis of scientific research is carried out correctly in terms of methodology. However, some doubts arise.

 - the analysis does not include a randomized, double-blind, controlled study (Davide Grassi, Giovambattista Desideri, Stefano Necozione, Paolo di Giosia, Remo Barnabei, Leen Allegaert, Herwig Bernaert, Claudio Ferri. Cocoa consumption dose-dependently improves flow-mediated dilation and arterial stiffness decreasing blood pressure in healthy individuals. J Hypertens. 2015 Feb; 33 (2): 294-303), on the basis of which the European Food Safety Authority issued a positive opinion on the possibility of using a health claim on a chocolate label: Cocoa flavanols help maintain the elasticity of blood vessels, which contributes to normal blood flow”.  The claim can be used only for cocoa beverages (with cocoa powder) or for dark chocolate which provide at least a daily intake of 200 mg of cocoa flavanols with a degree of polymerisation 1-10 (Commission Regulation (EU) No 851/2013 of 3 September 2013, authorizing certain health claims made on foods, other than those referring to the reduction of disease risk and to children's development and health and amending Regulation (EU) No 432/2012) Can the authors explain why they did not take this study into account?

- the title of the article concerns the health effects of chocolate, while among the 15 studies presented in the analysis, only 4 studies concern chocolate (as a product), and the rest concern cocoa bean ingredients, even in pharmaceutical form. Chocolate is a multi-ingredient product which, in the European Union, must meet the requirements of Directive 2000/36 / EC of the European Parliament and of the Council of 23 June 2000 relating to cocoa and chocolate products intended for human consumption. Therefore, the proposed title of the publication is very controversial

- in the Limitations of this study, the issue of short-term studies should be highlighted (6 out of 15 studies are 4-week studies and 2 others - 6 weeks). Information about the short duration of the research should also be added to the Conclusion of the article and Abstract.

During such a period, it is difficult to demonstrate both beneficial and unfavorable effects of consuming chocolate or cocoa ingredients. It would be more important from a health standpoint to know the effect of consuming chocolate regularly over a longer period of time.

The effect on body weight is of particular interest. The authors emphasize that studies in the context of anthropometric parameters are of very low quality, which means that one you should be cautious about the results obtained.

- in the Abstract, instead of "moderate-quality evidence" there should be "low-to-moderate-quality evidence", i.e. as in the results and in the conclusion

  - according to the methodology, triglyceride concentration is one of the parameters of the lipid profile. In the abstract of the article, the authors provide it separately (apart from the lipid profile), while in the summary it is included as lipid profile

Technical remarks:

- Table 1 should also contain the results of studies

- the numbering of figures should be corrected, because figure 8 appears twice

Author Response

Greetings Reviewer 2, thanks for your comments. Please refer to the attachment below. 
